# circEXOC5 promotes acute lung injury through the PTBP1/Skp2/Runx2 axis to activate autophagy

Pei Gao*, Beirong Wu*, Ying Ding, Bingru Yin, Haoxiang Gu

To understand the pathogenesis of acute lung injury (ALI), we focused on circEXOC5, a significantly up-regulated circular RNA in ALI. Using the in vivo cecal ligation and puncture (CLP)–induced ALI mouse model and in vitro LPS-challenged mouse pulmonary microvascular endothelial cell (MPVEC) model, we examined the impacts of knockdown circEXOC5 on lung injury, inflammation, and autophagy. The regulation between circEXOC5, polypyrimidine tract-binding protein 1 (PTBP1), S-phase kinase-associated protein 2 (Skp2), and Runt-related transcription factor 2 (Runx2) was investigated by combining RNA immunoprecipitation, qRT–PCR, mRNA stability, and ubiquitination assays. The significance of PTBP1 in circEXOC5-induced ALI phenotypes was examined both in vitro and in vivo. circEXOC5 was up-regulated and associated with increased inflammation and activated autophagy in cecal ligation and puncture–induced ALI lung tissues and LPS-challenged MPVECs. Through the interaction with PTBP1, circEXOC5 accelerated *Skp2* mRNA decay, an E3 ubiquitin ligase for Runx2, and therefore increased Runx2 expression. Functionally, overexpressing PTBP1 reversed shcircEXOC5-inhibited ALI, inflammation, or autophagy. The signaling cascade circEXOC5/PTBP1/Skp2/Runx2, by essentially regulating inflammation and autophagy in MPVECs, aggravates sepsis-induced ALI.

## Introduction

Acute lung injury (ALI) and its advanced form, acute respiratory distress syndrome (ARDS), are severe lung injuries frequently resulting from sepsis and presenting as life-threatening conditions to patients. The main histopathological features of ALI/ARDS include excessive neutrophil-derived inflammation, severe damages to alveolar epithelial cells and microvascular endothelial cells leading to the loss of alveolar-capillary barrier, and pulmonary edema (Johnson & Matthay, 2010; Mokra & Kosutova, 2015). To better understand the pathogenic mechanisms underlying the development of ALI, many experimental models have been established to mimic the various etiologies leading to ALI in humans, such as in vivo through cecal ligation and puncture (CLP) (Villar et al, 1994),

mechanical ventilation, pulmonary ischemia/reperfusion, and the administration of LPS, oleic acid, or bleomycin (Matute-Bello et al, 2008), and in vitro by challenging pulmonary epithelial cells, endothelial cells, or other cell types with LPS (Konter et al, 2012; Shao et al, 2017; Jiang et al, 2020a; Hui et al, 2021). With the help of various preclinical models, critical cell types (including epithelial cells, endothelial cells, and immune cells), various cellular processes (such as apoptosis, autophagy, and disruption of cellular junctions), and increasing bioactive molecules (including signaling molecules and cytokines) contributing to ALI development have been identified (Bhandari, 2008; Chopra et al, 2009; Fanelli & Ranieri, 2015). In particular, the relationship between autophagy and ALI is complicated by the causes, the variety of cell types, and the progressive stages of lung injury (Vishnupriya et al, 2020). Furthermore, the dismal prognosis suffered by ALI/ARDS patients still suggests the lack of comprehensive mechanistic understanding of the disease and the inefficiency in translating scientific findings into effective clinical approaches.

Circular RNAs (circRNAs) are a family of noncoding RNAs presenting crucial roles in regulating normal physiology and pathological developments. Several studies using systemic approaches and/or bioinformatics analysis all evidenced an altered expression profile of circRNAs in ALI (Bao et al, 2019; Li et al, 2019; Yuan et al, 2020; Jiang et al, 2020b), supporting their importance in ALI development. In contrast, the mechanistic understanding of specific circRNAs in ALI is still in its infancy (Zou et al, 2020; Yang et al, 2021) and largely remains to be revealed. To expand our understanding of circRNAs in ALI, we followed up on circEXOC5 (hsa_circ_0004399), one of the most significantly up-regulated circRNAs identified by microarray and verified by qRT–PCR in lung macrophages from CLP mice as compared to those from sham-operated mice (Bao et al, 2019). Functionally, one study reports the ferroptosis-promoting effect of circEXOC5 (Wang et al, 2022b) and another demonstrates its pyroptosis-promoting activity (Wang et al, 2022a) in the context of ALI. Mechanistically, circEXOC5 is shown to regulate the stability of downstream mRNAs and interact with polypyrimidine tract-binding protein 1 (PTBP1), a heterogeneous nuclear ribonucleoprotein (Wang et al, 2022b). However, little is known about the impacts and molecular mechanisms of circEXOC5 in autophagy, which may provide further breakthroughs to our understanding of the pathogenesis of ALI.

S-phase kinase-associated protein 2 (Skp2), a component of the E3 ubiquitin ligase SKP1-cullin-F-box (SCF), is well known as an

---

Department of Respiratory, Shanghai Children's Hospital, School of medicine, Shanghai Jiao Tong University, Shanghai, China

Correspondence: guxianghaoo4513@163.com
*Pei Gao and Beirong Wu are the first co-authors

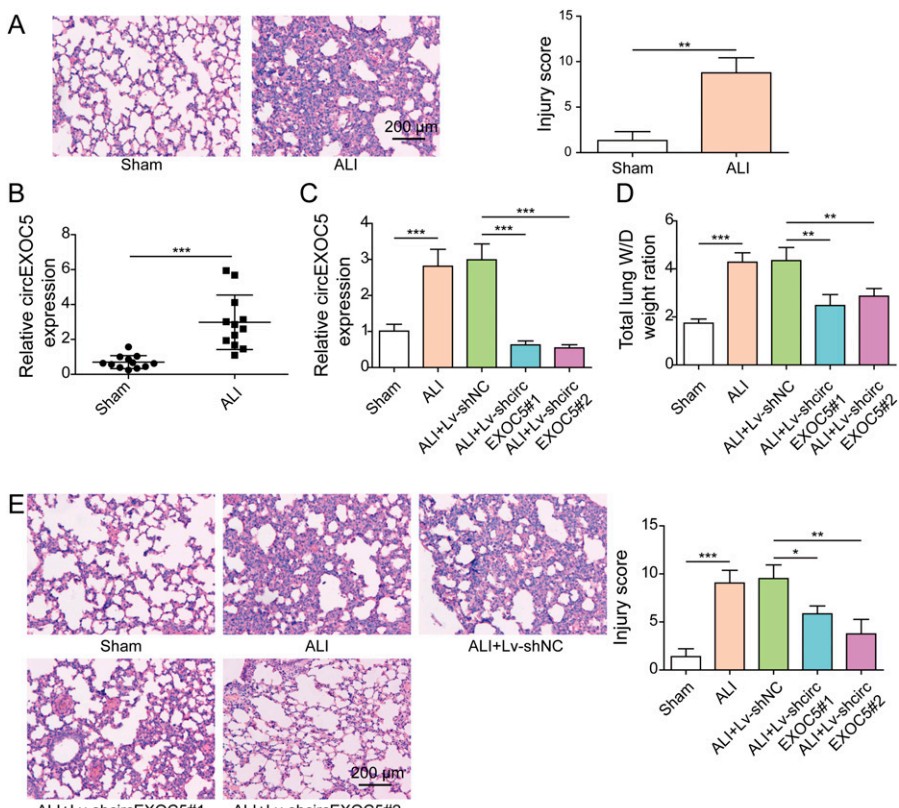

**Figure 1. Targeting circEXOC5 alleviated CLP-induced acute lung injury in vivo.**
**(A)** Representative HE images of lung tissues from CLP (acute lung injury) or sham-operated mice, with lung injury scored and compared between the groups. Scale bar: 200 μm. **(B)** Relative expression of circEXOC5 was examined by qRT–PCR in lung tissues from indicated groups. Mice were injected with Lv-shcircEXOC5 or Lv-shNC before the CLP procedure. **(C, D, E)** Relative expression of circEXOC5 (C), lung W/D weight ratio (D), and representative HE images and lung injury score (E) in lung tissues from indicated groups were examined. Scale bar: 200 μm. *P < 0.05, **P < 0.01, and ***P < 0.001.

essential regulator of cell-cycle progression and cancer development (Su et al, 2016). In addition, by promoting the ubiquitin degradation of Runt-related transcription factor 2 (Runx2), Skp2 inhibits osteogenesis (Thacker et al, 2016). Although the direct involvement of Skp2 in ALI is not clear, Runx2 has been identified as an important mediator of ALI (Zhou et al, 2018; Li et al, 2021). The mechanism leading to Runx2 up-regulation during ALI remains to be explored. Furthermore, we found that *Skp2* mRNA has a targeted binding site with PTBP1 through the analysis of the starBase database (http://starbase.sysu.edu.cn/). Therefore, we hypothesized that circEXOC5 decreased *Skp2* mRNA stability by targeting PTBP1, thereby inhibiting Skp2-mediated Runx2 ubiquitin degradation in ALI.

In this study, we combined in vivo CLP-induced ALI mouse model with the in vitro LPS-challenged mouse pulmonary microvascular endothelial cell (MPVEC) model, measured the expression of circEXOC5 in ALI, examined its functional impacts on inflammation and autophagy, and explored the underlying mechanisms, with specific focus on PTBP1, Skp2, and Runx2. This study reveals a novel mechanism by which circEXOC5 critically regulates the pathogenesis of ALI.

# Results

### Knocking down circEXOC5 alleviates CLP-induced ALI in vivo

To examine the potential involvement of circEXOC5 in ALI, we first established an in vivo CLP-induced ALI model. HE staining revealed characteristic ALI pathologies in CLP mice, including thickened alveolar walls, alveolar collapse, disruption of alveolar epithelium, and massive infiltration of red blood cells and inflammatory cells, which corresponded to highly elevated lung injury score (Fig 1A). Expression analysis of circEXOC5 by qRT–PCR (Fig 1B) revealed its significant up-regulation in ALI lungs, but not in lungs from sham mice. To assess its functional significance in ALI development, we injected lentivirus expressing two distinct shRNAs specifically targeting circEXOC5 (Lv-shcircEXOC5#1 and shcircEXOC5#2) into the tail vein of mice before performing the CLP procedure. When compared to mice injected with control lentivirus and CLP (ALI+Lv-shNC), we detected specific knockdown of circEXOC5 in lung tissues from ALI+Lv-shcircEXOC5#1 and from ALI+Lv-shcircEXOC5#2 mice (Fig 1C). Corresponding to the reduction in circEXOC5 in ALI+Lv-shcircEXOC5#1 and ALI+Lv-shcircEXOC5#2 mice, we noticed a significant alleviation of pulmonary edema (as indicated by lung W/D weight ratio; Fig 1D) and lung injury (as revealed by HE staining and lung injury score; Fig 1E), when compared to ALI or ALI+Lv-shNC mice. In collection, these data suggest that circEXOC5 is up-regulated and essentially controls the development of ALI.

### Knocking down circEXOC5 inhibits inflammation during ALI development

Inflammation plays a key role in the pathogenesis of ALI. To assess the importance of circEXOC5 in regulating inflammation, we first quantified cellular components in BALF. As shown in Fig 2A, the number of total cells, neutrophils, and macrophages was all significantly increased in BALF from ALI mice, when compared to sham

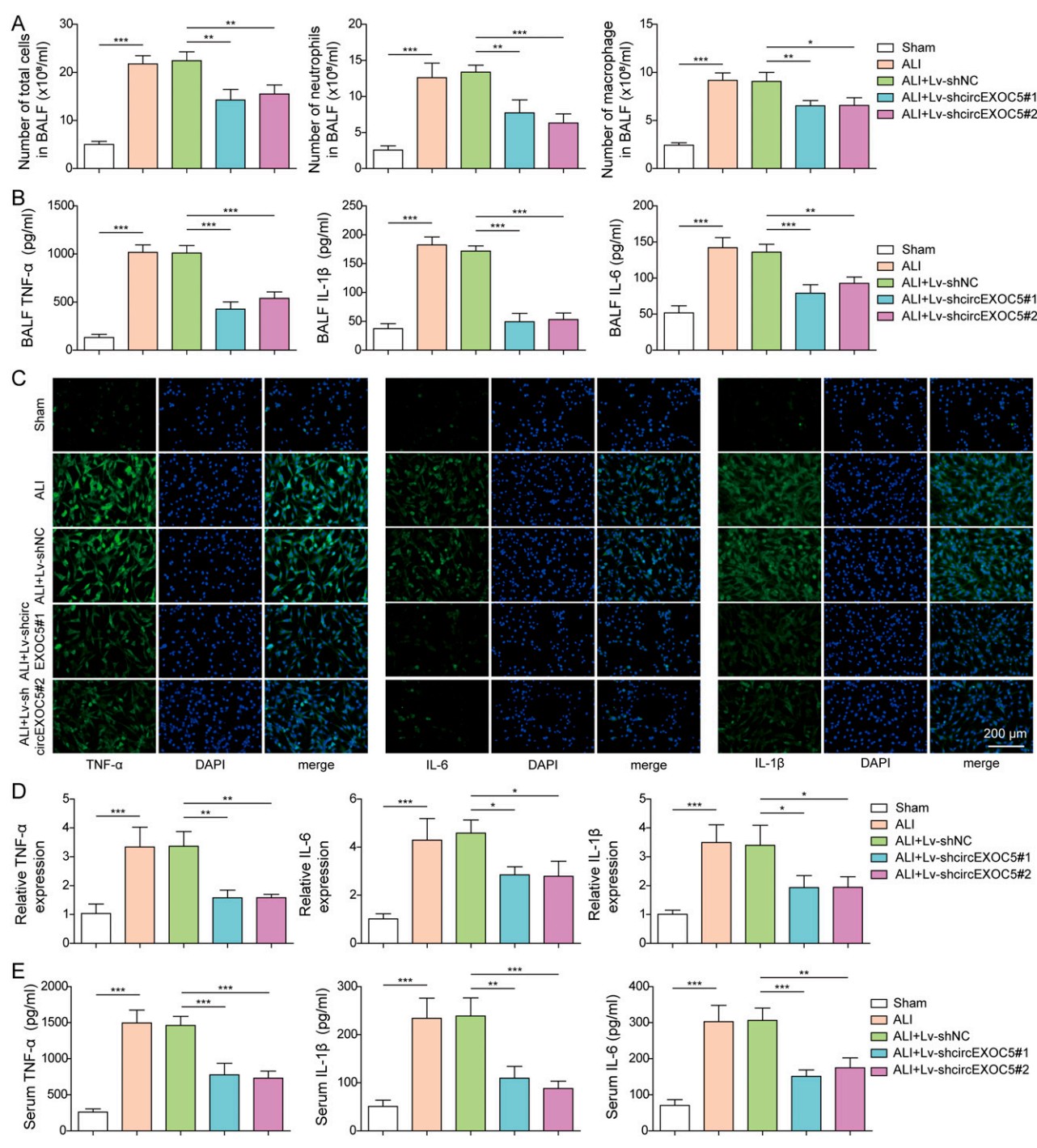

**Figure 2. Knockdown of circEXOC5 reduced inflammation in CLP-induced acute lung injury.**
**(A)** Numbers of total cells, neutrophils, and macrophages were quantified. **(B)** Production of TNF-α, IL-6, and IL-1β was measured by ELISA in BALF. **(C, D)** Expressions of TNF-α, IL-6, and IL-1β were determined by immunofluorescence (C) and qRT–PCR (D) in isolated macrophages from BALF from indicated groups. **(E)** Production of TNF-α, IL-6, and IL-1β was measured by ELISA in mouse serum. *P < 0.05, **P < 0.01, and ***P < 0.001.

mice, implying elevated inflammation. In contrast, when compared to BALF from ALI or ALI+Lv-shNC mice, BALF from ALI+Lv-shcircEXOC5#1 or ALI+Lv-shcircEXOC5#2 mice contained a significantly reduced number of all three cell populations. The reduction in inflammatory cells by knocking down circEXOC5 was also associated

with their decreased levels of pro-inflammatory cytokines including TNF-α, IL-6, and IL-1β in BALF (Fig 2B), their expressions from isolated macrophages from BALF (Fig 2C and D), and their systemic levels detected in serum (Fig 2E), suggesting that knocking down circEXOC5 was sufficient to inhibit inflammation during ALI development.

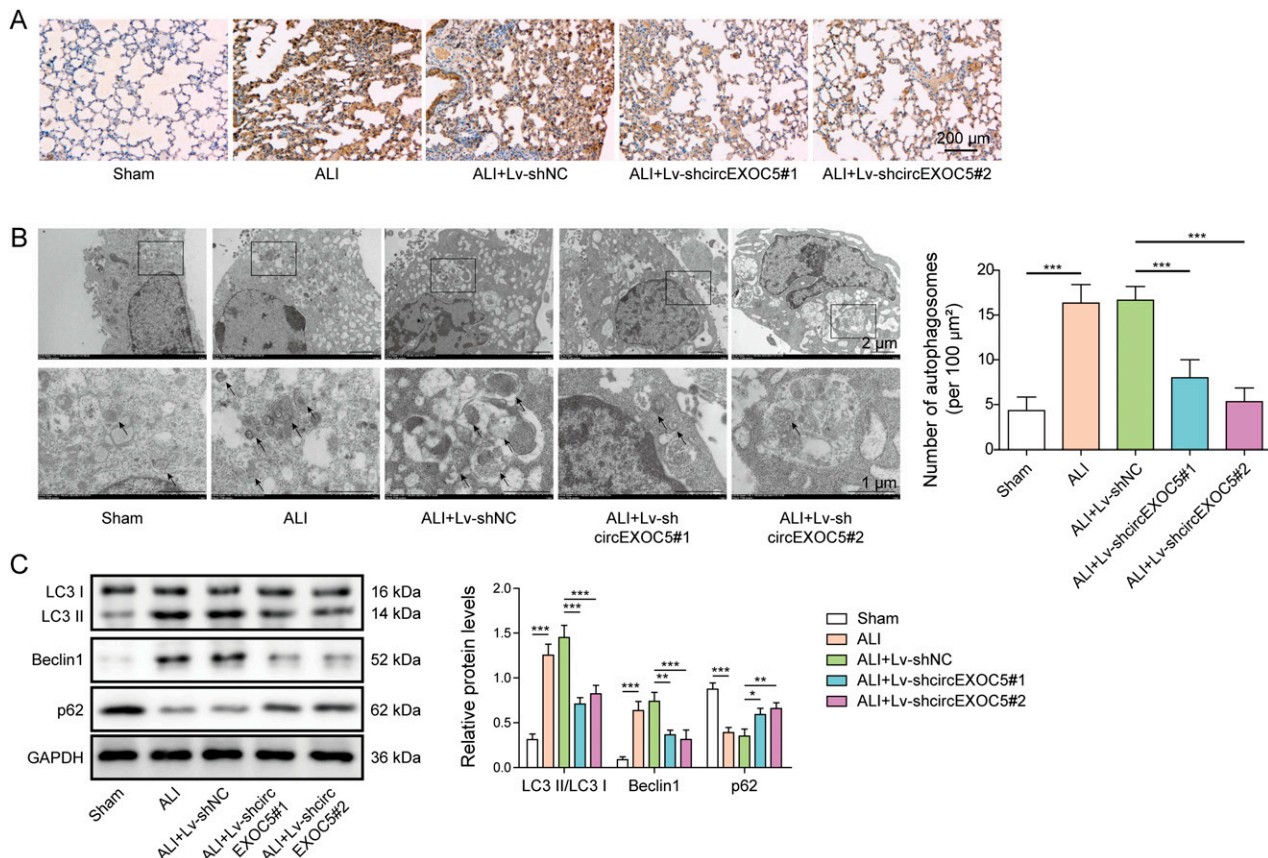

**Figure 3.  Silencing circEXOC5 suppressed the autophagy activation of CLP-induced acute lung injury.**
**(A)** In lung tissues from indicated groups, the expression of LC3 was examined by immunohistochemical analysis (A). Scale bar: 200 μm. **(B, C)** Autophagosomes were observed under a transmission electron microscope, with the number counted (B), and expressions of LC3I, LC3II, Beclin1, p62, and GAPDH (internal control) were measured by Western blotting (C). *P < 0.05, **P < 0.01, and ***P < 0.001.

## Targeting circEXOC5 suppresses autophagy in ALI

An earlier study showed that autophagy activation is a mechanism leading to inflammation and ALI damages (Zhao et al, 2019). To understand whether shcircEXOC5 benefits ALI by inhibiting autophagy, we compared autophagy in lung tissues from sham mice, ALI, ALI+Lv-shNC, ALI+Lv-shcircEXOC5#1, and ALI+Lv-shcircEXOC5#2 mice. Consistent with the earlier study (Zhao et al, 2019), we observed robust increases in LC3 expression by immunohistochemical analysis (Fig 3A), in the number of autophagosomes by transmission electron microscope (TEM) (Fig 3B), and in the levels of LC3 and Beclin1 (Fig 3C), while reduced expression of p62 (Fig 3C), in ALI or ALI+Lv-shNC lungs, when compared to sham group lungs. However, all these changes witnessed in ALI mice were significantly dampened in ALI + Lv-shcircEXOC5#1 and in ALI+Lv-shcircEXOC5#2 mice, suggesting the essential role of circEXOC in activating autophagy and inducing ALI.

## Silencing of circEXOC5 inhibits LPS-induced autophagy and inflammation in MPVECs

To understand the molecular mechanisms underlining the control of circEXOC5 on autophagy and inflammation, we used LPS-treated MPVECs as in vitro model of ALI and silenced the endogenous expression of circEXOC5 with Lv-shcircEXOC5#1 or Lv-shcircEXOC5#2. To examine the importance of autophagy, we used 3-MA (a widely used autophagy inhibitor through its inhibition of PI3K), bafilomycin A1 (a potent and selective autophagy inhibitor via blocking vacuolar H⁺-ATPases), and rapamycin (an autophagy inducer by inhibiting mTORC1). Consistent with the in vivo findings, we found that LPS alone or LPS + Lv-shNC was sufficient to increase the production of pro-inflammatory TNF-α, IL-6, and IL-1β (Fig 4A), promote cell death (Fig 4B), elevate the expressions of LC3 and Beclin1, whereas reduce that of p62 (Fig 4C and D) from MPVECs. Silencing circEXOC5 (LPS + Lv-shcircEXOC5#1 or LPS + Lv-shcircEXOC5#2) significantly inhibited all these LPS-induced changes. Simultaneously silencing circEXOC5 and blocking autophagy with 3-MA (LPS + Lv-shcircEXOC5#1 + 3-MA) or with a more specific autophagy inhibitor, bafilomycin A1 (LPS + Lv-shcircEXOC5#1 + bafilomycin A1) further aggravated and completely abolished, while silencing circEXOC5 and activating autophagy with rapamycin (LPS + Lv-shcircEXOC5#1 + rapamycin) recovered all LPS-induced changes. 3-MA and rapamycin are broad-spectrum regulators controlling both autophagy-dependent and autophagy-independent pathways. However, the comparable data achieved by 3-MA and bafilomycin A1 strongly support the involvement of autophagy in this process. Taken together, these data suggest that circEXOC5, by inducing autophagy, promotes LPS-induced inflammation of MPVECs.

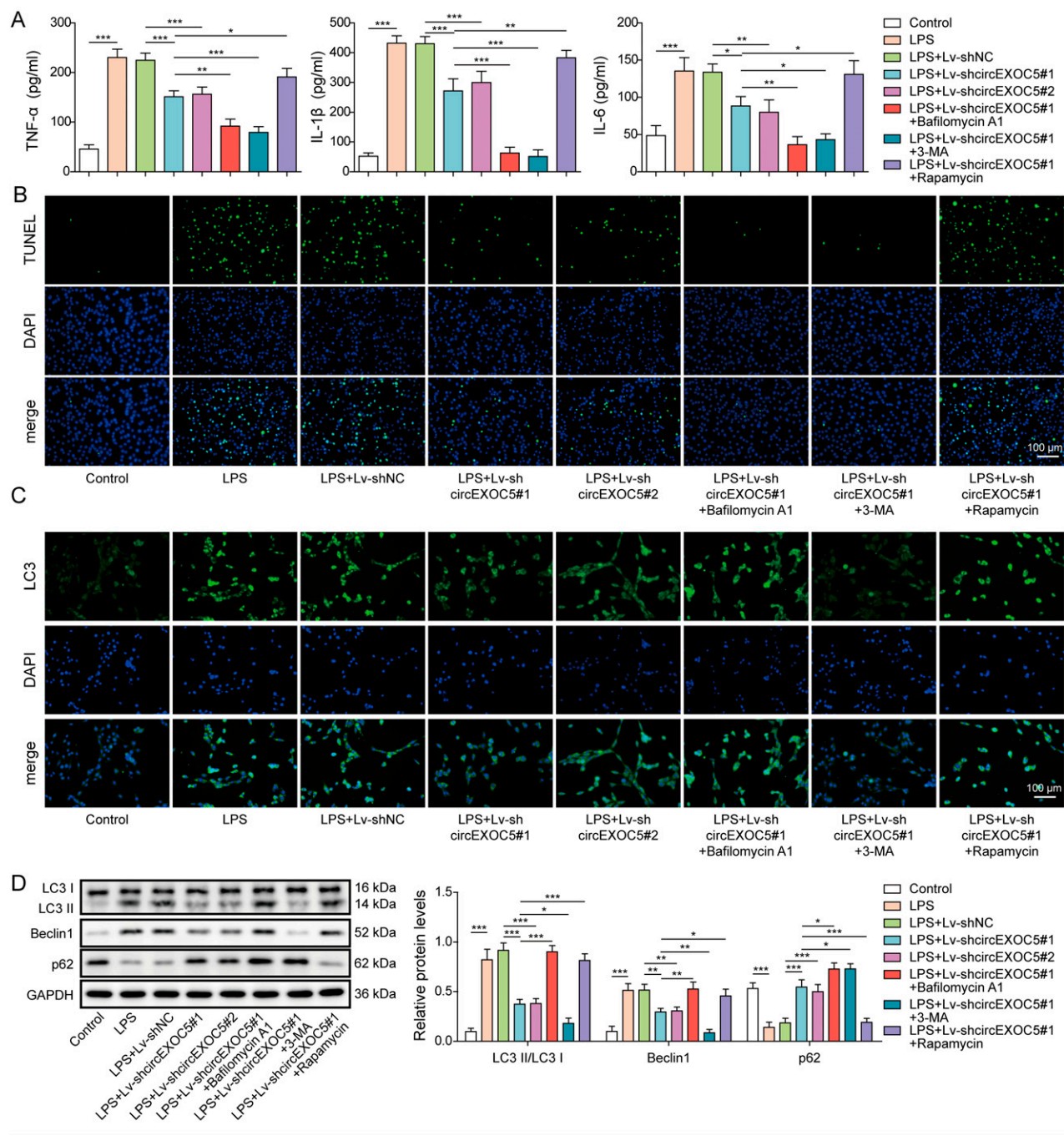

**Figure 4. Silencing of circEXOC5 inhibited LPS-induced autophagy and inflammation in MPVECs.**
MPVECs were stably transfected without or with Lv-shNC or Lv-shcircEXOC5 and treated with PBS (control), LPS, LPS + 3-MA, or LPS + rapamycin. **(A)** Production of TNF-α, IL-6, and IL-1β was measured by ELISA. **(B)** Cell death was examined by TUNEL assay. **(C)** Expression of LC3 (green) was detected by immunofluorescence. Cell nuclei were stained with DAPI (blue). **(D)** Expressions of LC3I, LC3II, Beclin1, p62, and GAPDH (internal control) were measured by Western blotting. *P < 0.05, **P < 0.01, and ***P < 0.001.

## PTBP1 directly interacts with circEXOC5

To examine the potential involvement of PTBP1 in circEXOC5-regulated sepsis injury, we first measured its expression on mRNA and protein levels in lung tissues from mice undergoing different treatments as shown in Fig 1. We found that both *PTBP1* mRNA (Fig 5A) and protein (Fig 5B and C) levels markedly increased in ALI and ALI + Lv-shNC lungs but failed to increase in ALI + Lv-shcircEXOC5#1 or ALI+Lv-shcircEXOC5#2 lungs, implying the up-regulation of PTBP1 by circEXOC5. The regulation between circEXOC5 and *PTBP1* was further corroborated by the positive correlation between these two molecules in ALI lungs (Fig 5D). In MPVECs, silencing circEXOC5 with

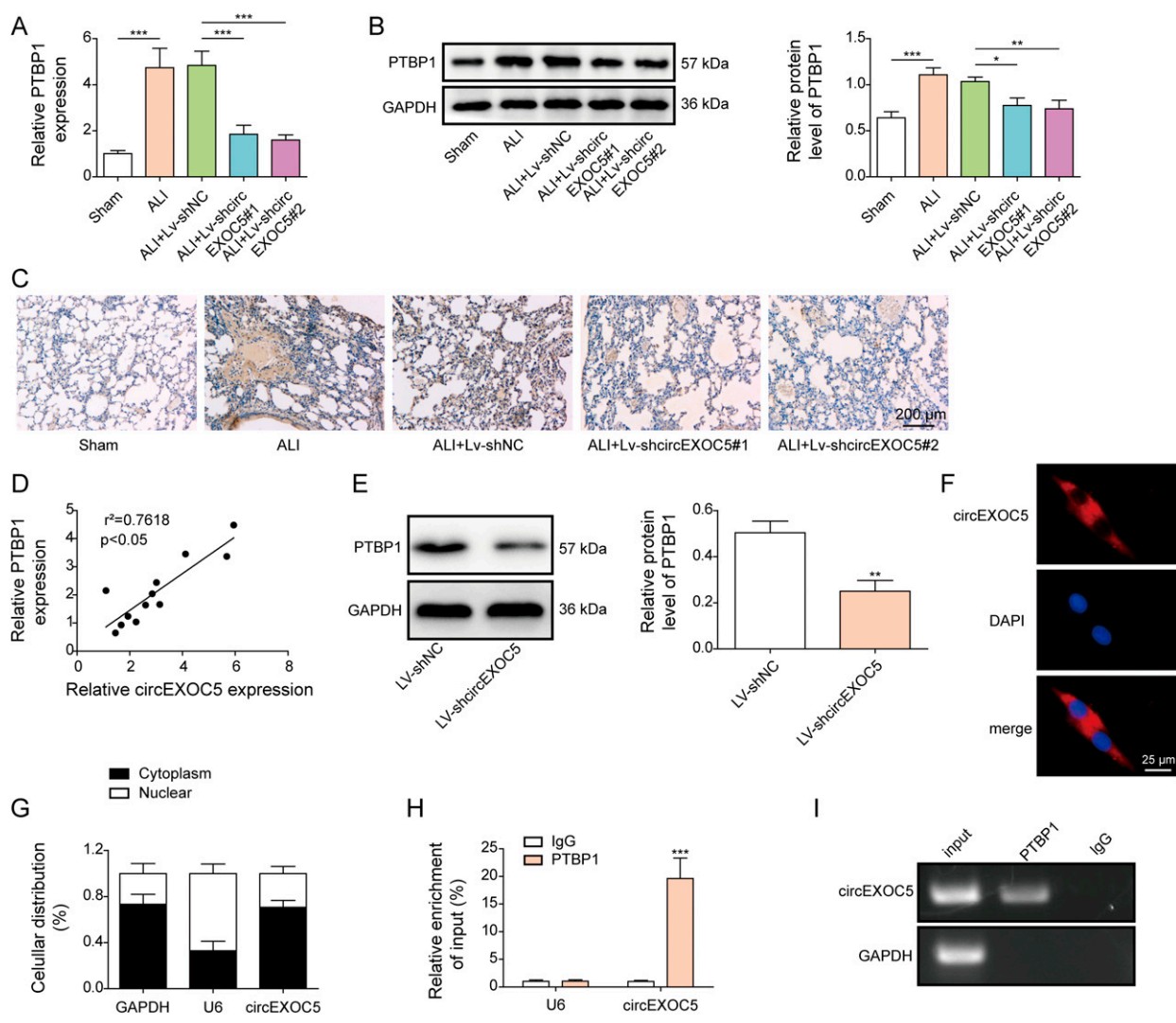

**Figure 5. PTBP1 directly interacts with circEXOC5.**
**(A, B, C)** Expression of PTBP1 mRNA and protein in lung tissues of indicated mouse groups was detected by qRT–PCR (A), Western blotting (B), and immunohistochemical analysis (C). **(D)** Expression correlation between circEXOC5 and *PTBP1* in lung tissues from mice with acute lung injury was examined by the Pearson correlation analysis. **(E)** PTBP1 was detected by Western blotting in Lv-shNC and Lv-shcircEXOC5 MPVECs. **(F)** Localization of circEXOC5 in MPVECs was examined by RNA-FISH (red for circEXOC5 and blue for DAPI) for indicated subcellular fractions. Scale bar: 25 *μ*m. **(H, I)** Interaction between circEXOC5 and PTBP1 was detected by RIP assay. **P < 0.01 and ***P < 0.001.

Lv-shcircEXOC5#1 or Lv-shcircEXOC5#2 was sufficient to down-regulate PTBP1 expression (Fig 5E). RNA-FISH (Fig 5F) and cell fractionation (Fig 5G) assays detected circEXOC5 in both nucleus and cytoplasm, but preferentially in the cytoplasm. To further explore the mechanism by which circEXOC5 positively promotes PTBP1, their interaction was further confirmed using RIP assay (Fig 5H and I). In collection, these data suggest that PTBP1 directly interacts with circEXOC5.

## Overexpressing PTBP1 reverses shcircEXOC5-induced protection of MPVECs

To further study the positive regulation between circEXOC5 and PTBP1, we overexpressed either empty vector (OE-NC) or PTBP1 (OE-PTBP1) in Lv-shcircEXOC5 MPVECs, before treating these cells with LPS to induce ALI damages. As shown Fig 6A, the transfection of OE-PTBP1

could significantly increase the mRNA and protein expressions of PTBP1 in MPVECs. OE-NC did not significantly alter the phenotypes of Lv-shcircEXOC5 (LPS+Lv-shcircEXOC5+OE-NC) cells, including the production of inflammatory cytokines TNF-α, IL-6, and IL-1β (Fig 6B), cell death (Fig 6C), and expressions of LC3, Beclin1, and p62 (Fig 6D and E). In contrast, overexpressing PTBP1 in LPS+Lv-shcircEXOC5+OE-PTBP1 cells completely reversed all phenotypes induced by silencing circEXOC5, suggesting that PTBP1 acts downstream of circEXOC5 and critically mediates impacts of the latter on LPS-challenged MPVECs.

## circEXOC5-PTBP1 interaction reduces *Skp2* mRNA stability, whereas the latter mediates Runx2 ubiquitination

Analysis using the starBase platform (http://starbase.sysu.edu.cn/) revealed the potential interaction between PTBP1 and Skp2 mRNA,

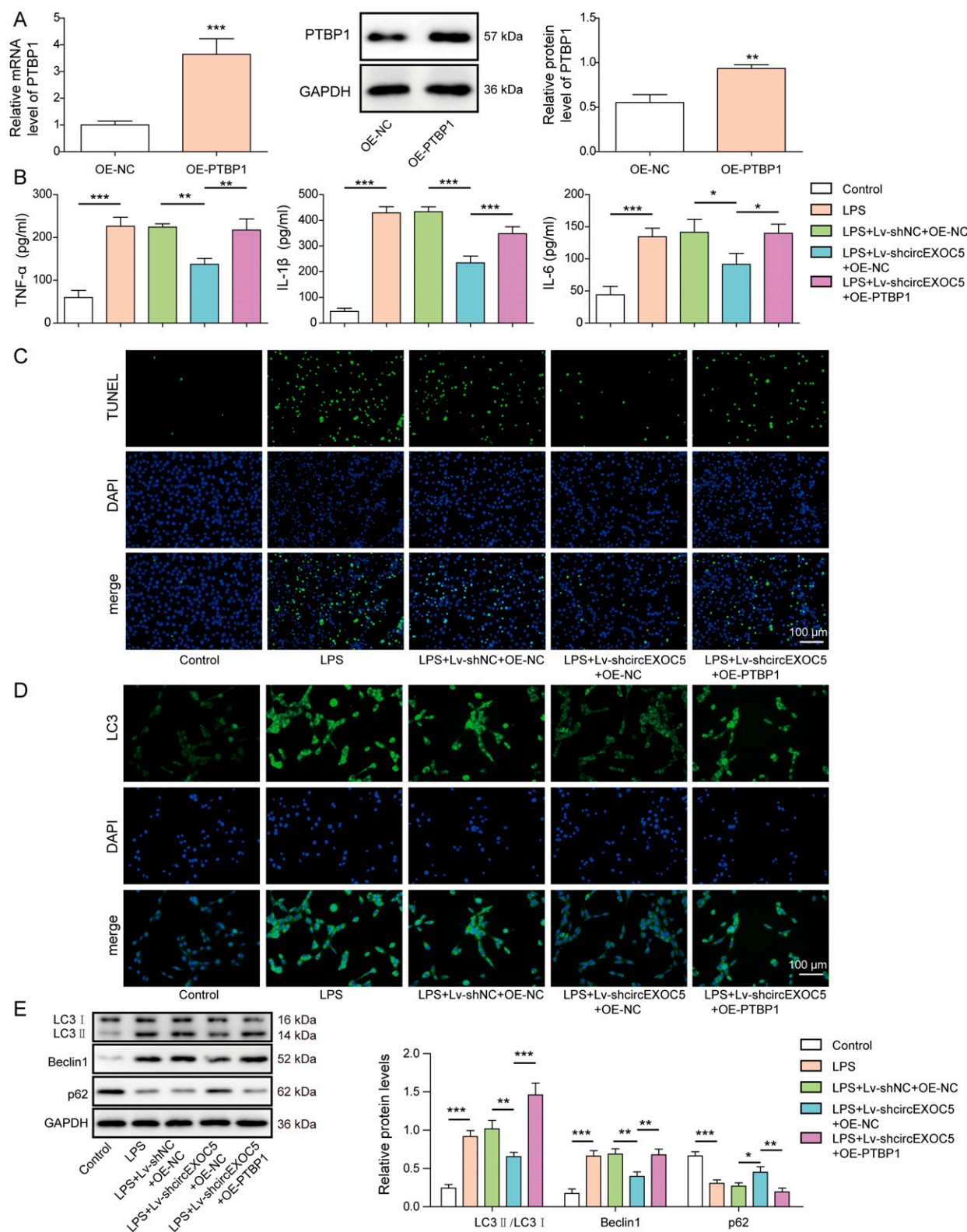

**Figure 6. Overexpressing PTBP1 reversed the protective effect of silencing circEXOC5 on MPVECs.**
**(A)** Expression of PTBP1 mRNA and protein in MPVECs transfected with OE-NC and OE-PTBP1 was detected by qRT–PCR and Western blotting. MPVECs were stably transfected without or with Lv-shNC, Lv-shcircEXOC5, OE-NC, or OE-PTBP1 before being treated with PBS (control) or LPS. **(B)** Production of TNF-α, IL-6, and IL-1β was measured by ELISA. **(C)** Cell death was examined by TUNEL assay. **(D)** Expression of LC3 (green) was detected by immunofluorescence. Cell nuclei were stained with DAPI (blue). **(E)** Expressions of LC3I, LC3II, Beclin1, p62, and GAPDH (internal control) were measured by Western blotting. *P < 0.05, **P < 0.01, and ***P < 0.001.

although Skp2 is an E3 ubiquitin ligase that promotes the proteasomal degradation of Runx2 (Thacker et al, 2016), the latter shown to promote ALI (Wu et al, 2018; Zhao et al, 2019). Therefore, we examined the potential involvement of Skp2 in the regulation between circEXOC5 and PTBP1 in Runx2. Expression analysis by qRT–PCR showed that knocking down either circEXOC5 (Fig S1A) or PTBP1 (Fig S1B) was sufficient to up-regulate Skp2 mRNA, but overexpressing PTBP1 reversed the *Skp2* up-regulation induced by knocking down circEXOC5 (Fig S1C), suggestive of the importance of circEXOC5-PTBP1 interaction. When focusing on Skp2 mRNA stability in MPVECs, we noted that Lv-shcircEXOC5 promoted the stability of Skp2 mRNA, whereas overexpressing PTBP1 in Lv-shcircEXOC5 cells abolished the stabilizing effect (Fig S1D). RIP assay (Fig S1E and F) further verified the interaction between PTBP1 and *Skp2* mRNA. On the protein level, we showed that Skp2 was up-regulated, whereas Runx2 was down-regulated in Lv-shcircEXOC5 cells, when compared to Lv-shNC cells. Overexpressing Skp2 in both Lv-shNC and Lv-shcircEXOC5 cells further reduced the protein levels of Runx2 (Fig S1G). Co-IP assay detected the ubiquitination of Runx2 in shNC MPVECs, which was significantly reduced in shSkp2 MPVECs (Fig S1H). Together, these data demonstrated the significance of circEXOC5-PTBP1 interaction, by destabilizing Skp2 mRNA, suppresses ubiquitination-mediated degradation of Runx2.

### circEXOC5/PTBP1 signaling is essential for ALI development in vivo

To assess the in vivo significance of PTBP1 in circEXOC5-regulated ALI development, we injected Lv-shcircEXOC5 or Lv-shcircEXOC5+-OE-PTBP1 into mice before performing the CLP procedure. Consistent with earlier findings, Lv-shcircEXOC5 significantly alleviated CLP-induced ALI (as represented by HE staining and injury score; Fig 7A), reduced inflammation (as indicated by the reduced number of total cells, neutrophils, and macrophages in Fig 7B, and the production of pro-inflammatory TNF-α, IL-6, and IL-1β in Fig 7C), and inhibited autophagy (as suggested by a reduced number of autophagosomes in Fig 7D). These changes were associated with a reduced expression of Runx2 in the lung tissue (Fig 7E). However, overexpressing PTBP1 together with Lv-shcircEXOC5 completely abolished the preventive benefits or the impacts of the latter on inflammation and autophagy, suggesting the essential role of PTBP1 in mediating the in vivo effects of circEXOC5 on ALI.

## Discussion

ALI/ARDS accounts for ~10% ICU admissions and is associated with a striking mortality rate from nearly 30% for those with mild cases to more than 46% in severe cases (Bellani et al, 2016). Clearly, it is imperative to develop more effective strategies to manage/treat patients with ALI/ARDS, which largely depends on our understanding of ALI pathogenesis. In this study, we identified a novel circRNA critical for the development of ALI. Using a CLP-induced ALI mouse model, we showed that circEXOC5 was significantly up-regulated in ALI and associated with the presence of inflammation and autophagy in lung tissues. Functionally, knocking down

circEXOC5 was sufficient to alleviate ALI-related injury, inflammation, or autophagy. Mechanistically, circEXOC5, through the up-regulation of and the interaction with PTBP1, stimulates the mRNA degradation of *Skp2* and stabilizes Runx2 to promote autophagy and ALI development.

Autophagy is a multi-step biological process by which cells package misfolded proteins, damaged organelles, and/or invading pathogens into autophagosomes, fuse them with lysosomes to form autolysosomes, and degrade the contents to recycle energy and cell nutrients (Khandia et al, 2019). Although the presence of autophagosomes is frequently associated with cell death, autophagy is generally considered a mechanism for cell protection, instead of a cause for cell death (Kroemer & Levine, 2008). During the development of ALI induced by various causes, the activation of autophagy, as evidenced by the increased formation of autophagosomes and the up-regulation of autophagy markers such as LC3II and Beclin1, is ubiquitously detected in various cell types; however, its functional role is still under debate. Some studies suggest elevated autophagy is a maladaptive response caused by incomplete autophagy activation, and therefore, completing or enhancing autophagy will alleviate ALI and improve the prognosis (Lo et al, 2013; Lin et al, 2014; Zhao et al, 2019; Nosaka et al, 2020; Peng et al, 2021; Zhang et al, 2021). On the contrary, other studies indicate activated autophagy is a pathogenic factor leading to ALI, and thus, inhibiting autophagy benefits the treatment (Sun et al, 2012; Hu et al, 2014; Zhu et al, 2017; Slavin et al, 2018; Chen et al, 2020; Li et al, 2020). In this study, we detected activated autophagy in CLP-induced ALI damages, as represented by the increased number of autophagosomes under TEM, the up-regulated expression and co-localization of LC3II and Beclin1, and the down-regulation of p62. These changes were concomitant with exacerbated inflammation and lung injury, supporting the involvement of autophagy in ALI development. When knocking down circEXOC5 systematically, we observed the simultaneous alleviation of autophagy, inflammation, and lung injury, suggesting preventive benefits of targeting autophagy in ALI. Furthermore, using LPS-challenged MPVECs as the in vitro model, we evidenced the similar benefits of silencing circEXOC5 in suppressing autophagy and inflammation, and these benefits were further boosted by 3-MA or bafilomycin A1, autophagy inhibitors, but reversed with rapamycin, an autophagy inducer. Therefore, this study corroborates that autophagy is a detrimental factor contributing to the pathogenesis of sepsis-induced ALI. Although we only focused on microvascular endothelial cells in in vitro studies, it would be interesting to examine whether and how circEXOC5 regulates autophagy in other cell types important for ALI, such as alveolar epithelial cells and immune cells, and whether such regulation is beneficial or detrimental to their functions.

Cumulative evidence suggests the importance of circRNAs in autophagy. For example, circHIPK3, by sponging miR-20b-5p and up-regulating ATG7, promotes cardiomyocyte autophagy and apoptosis during ischemia/perfusion injury (Qiu et al, 2021). Hsa_circ_0003489, by sponging miR-874-3p and increasing HDAC1 expression, stimulates autophagy and contributes to the drug resistance of myeloma cells (Tian et al, 2021). In comparison, fewer studies have examined the involvement of specific circRNAs in ALI. Zou et al (2020) showed that the protective effects of P2X$_7$ receptor antagonist against sepsis-induced ALI were mediated through the

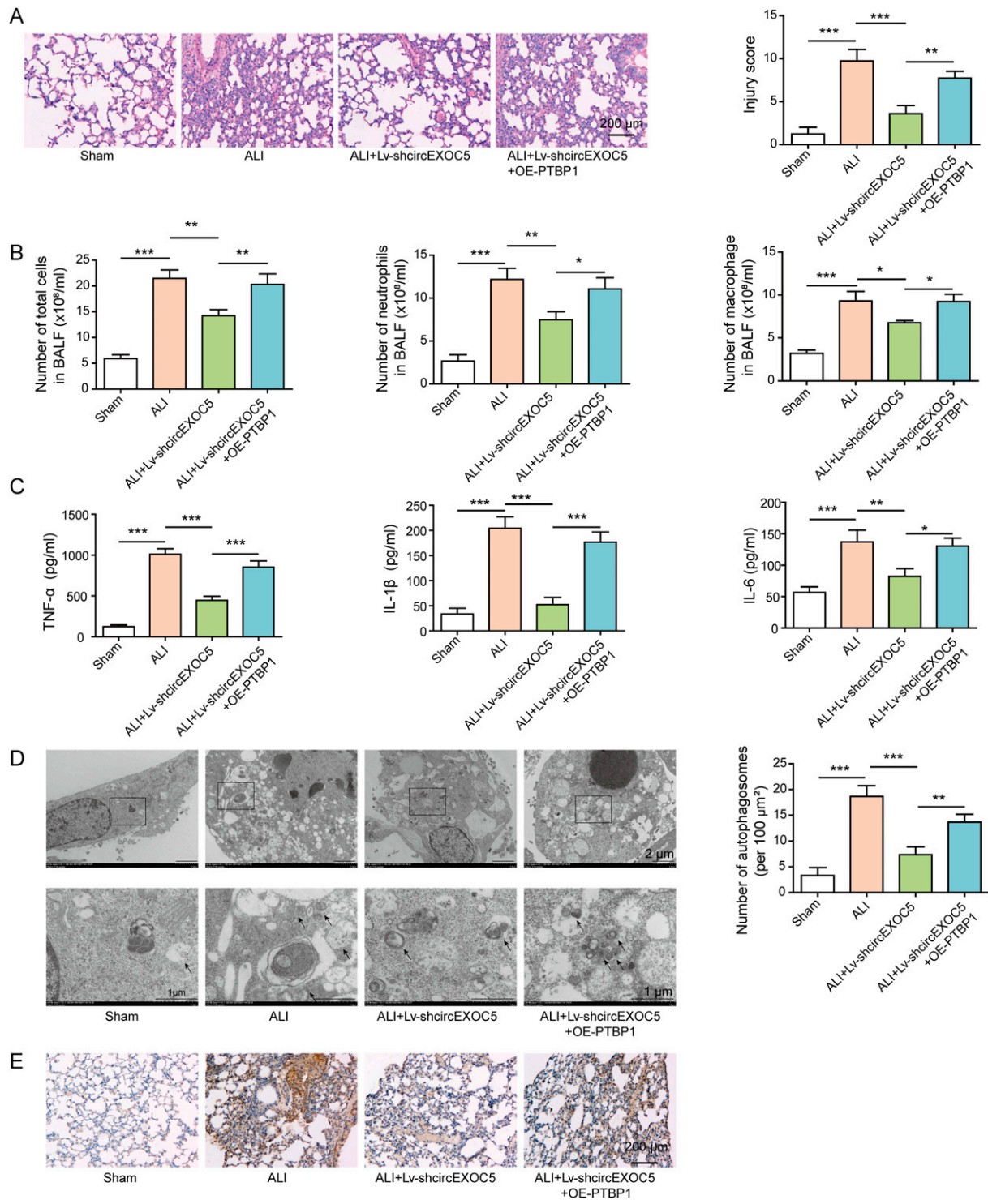

**Figure 7. Overexpressing PTBP1 abolished the in vivo preventive benefits of silencing circEXOC5 on acute lung injury.**
Mice were injected with Lv-shcircEXOC5 or Lv-shcircEXOC5 + OE-PTBP1 before the CLP procedure. **(A)** Histology of lung tissues from indicated groups was examined by HE staining and scored for lung injury. Scale bar: 200 μm. **(B)** Numbers of total cells, neutrophils, and macrophages in BALF were quantified. **(C)** Production of TNF-α, IL-6, and IL-1β in BALF was measured by ELISA. **(D)** Autophagosomes in lung tissues were observed under a transmission electron microscope, with the number counted from indicated groups. Scale bar: 2 or 1 μm. **(E)** Runx2 expression was detected by immunohistochemical analysis in indicated lung tissues. Scale bar: 200 μm. *$P < 0.05$, **$P < 0.01$, and ***$P < 0.001$.

up-regulation of circ_0001679 and circ_0001212, and their mRNA targets (Zou et al, 2020). Yang et al (2021) reported that circ_0054633 was up-regulated, activated the NF-κB signaling, and promoted the proliferation and inflammation of MPVECs in LPS-induced ALI (Yang et al, 2021). In this study, we followed up on a novel circRNA, circEXOC5, that was shown to be significantly up-regulated in macrophages from CLP-induced ALI (Bao et al, 2019), and demonstrated its significance in promoting autophagy, inflammation, and ALI, both in CLP-induced ALI model and LPS-challenged MPVEC model. To our knowledge, circEXOC5 is the first circular RNA linking autophagy and ALI. In addition, unlike most studies revealing circRNA-miRNA-mRNA network as the key mechanism for circRNA function, here we showed that circEXOC5 up-regulated Runx2 to stimulate autophagy. However, this up-regulation was not mediated through a miRNA, but through the up-regulation of and the interaction with PTBP1 and by accelerating the degradation of *Skp2* mRNA.

As an RBP, PTBP1 is known for regulating gene expression on the post-transcriptional level, including mRNA splicing, stability, translation, and localization (Zhu et al, 2020). Sun et al (2019) showed that circMYBL2 facilitated the binding of PTBP1 to *FLT3* mRNA, enhanced FLT3 translation of FL, and promoted leukemia progression (Sun et al, 2019). Wang et al (2019) reported that circFOXP1, through the interaction with PTPB1, stabilized *PKLR* mRNA (Wang et al, 2019). Here, we added the new finding that circEXOC5-PTBP1 interaction accelerated the decay of *Skp2* mRNA, which then indirectly protected Runx2 from ubiquitin-mediated degradation.

Runx2 is a transcription factor crucially controlling bone development (Kim et al, 2020) and also a facilitator of autophagy in different disease paradigms, such as osteogenesis, breast cancer, and vascular calcification (Yao et al, 2017; Qin & Cai, 2018; Tandon et al, 2018). Several recent studies examined the involvement of Runx2 in ALI. Li et al (2013, 2019, 2020, 2021) showed that Runx2 contributed to LPS-induced inflammatory injury, and by targeting Runx2, miR-30a-5p ameliorated LPS-induced ALI (Yang et al, 2021). Similarly, miR-218 alleviated sepsis-induced ALI by inhibiting Runx2-related inflammation (Zhou et al, 2018). Although these studies support the pathogenic impacts of Runx2 on ALI, it is not clear whether the pathogenic effects of Runx2 involve its regulation on autophagy. In the present study, we showed that overexpressing Runx2 could reverse the inhibition of silencing circEXOC5 on inflammation and autophagy, which translated to the loss of preventive benefits on ALI from silencing circEXOC5, supporting the importance of autophagy in the pathogenesis of ALI. circEXOC5, by up-regulating and interacting with PTBP1, promoted the degradation of *Skp2*. Skp2 is an E3 ubiquitin ligase of Runx2, whereas Runx2 was shown in earlier studies to down-regulate LC3, mostly in osteoblasts (Qin & Cai, 2018; Ren et al, 2022) and confirmed in this study. Therefore, knocking down circEXOC5 was sufficient to suppress the LPS-induced up-regulation of LC3. Because PTBP1 acts downstream of circEXOC5, overexpressing PTBP1 was able to reverse the inhibitory effect of shcircEXOC5 on LC3.

Despite the significant findings from this study, we realized that it is associated with some limitations. First, we only focused on male mice in this study, and thus, the mechanistic and functional significance of the circEXOC5/PTBP1/Skp2/Runx2 axis in female mice should be further verified. Second, we mainly studied the lung tissue here and did not examine the potential effect of knocking down circEXOC5 on other organs, which should be explored in the future to verify the safety profile of silencing circEXOC5. Third, although ALI was alleviated by silencing circEXOC5 before the CLP procedure in vivo or before the LPS treatment in vitro, our findings imply the preventive but not the therapeutic benefits of silencing circEXOC5. Future studies should focus more on the therapeutic significance.

In summary, this study reveals a novel pathogenic circRNA, circEXOC5 in ALI development. We not only demonstrate the up-regulation of circEXOC5 in ALI and its essential role in modulating inflammation and autophagy in both CLP lung tissues and LPS-challenged MPVECs, but also reveal the circEXOC5/PTBP1/Skp2/Runx2 axis in this process. This study provides preclinical evidence that altering this axis may significantly impact the pathogenesis of ALI. Consistently, a recent study by Wang et al (2022b) reveals that circEXOC5 interacts with PTBP1, promotes ferroptosis, and aggravates sepsis-induced ALI (Wang et al, 2022b). Therefore, in addition to regulating autophagy, the circEXOC5–PTBP1 interaction may act on multiple pathogenic mechanisms of ALI, which should be further investigated systemically in the future.

# Materials and Methods

## CLP-induced ALI model

All animal protocols were reviewed and approved by the Institutional Animal Care and Use Committee of Shanghai Children's Hospital. Male C57BL/6J mice of 8–9 wk of age were purchased from Shanghai Laboratory Animal Center. CLP was performed as described previously (Rittirsch et al, 2009). In brief, mice were anesthetized with a combination of ketamine (100 mg/kg; Fujian Gutian Pharmaceutical) and xylazine (10 mg/kg; Shengda Animal Medicine). A 1-cm incision was then made along the midline, and the cecum was exposed. A ligation was made on the middle of the cecum using a 3/0 silk suture, and two punctures were generated with a 21-gauge needle to the distal of the ligation. After extruding a small amount of stool, the cecum was returned and the abdomen was closed. Sham-operated mice with no CLP procedure were used as the negative control. For fluid resuscitation, all mice received a subcutaneous injection of 1 ml of 37°C normal saline immediately after the procedure. To alter the expression of circEXOC5 and/or PTBP1, lentivirus expressing control shRNA (Lv-shNC), circEXOC5 shRNA (Lv-shcircEXOC5), overexpressing control (OE-NC), and/or OE-PTBP1 (GenePharma) were injected intravenously into the tail vein of mice ($1 \times 10^8$ MOI in 200 µl of saline; n = 12/group) at 1 wk before the CLP procedure. All mice were closely monitored daily and euthanized at 72 h after the surgery or if they appeared moribund.

## Histopathological analysis

Upon euthanasia, lung tissues were prepared into formalin-fixed, paraffin-embedded sections (4 µm in thickness) and stained with hematoxylin and eosin (HE) following the protocol from an HE

staining kit (Vector Labs). To score lung injury, HE-stained lung tissues were blindly analyzed by two pathologists for the following four features: alveolar membrane thickening, capillary congestion, intra-alveolar hemorrhage, and neutrophil infiltration.

Immunohistochemical staining of microtubule-associated protein 1A/1B-light chain 3 (LC3), PTBP1, or Runx2 was performed as described earlier (Rosenfeldt et al, 2012). In short, tissue sections were deparaffinized in xylene and rehydrated in diluted series of ethanol. After antigen retrieval in boiling buffer (10 mM Tris–EDTA, pH 9) for 10 min, tissue sections were blocked and then incubated with anti-LC3 (1:100; ab63817), anti-PTBP1 (1:100; ab133734), or anti-Runx2 antibody (1:500; ab192256; all from Abcam) followed by HRP-conjugated secondary antibody (Abcam). The signal was developed using a DAB substrate (Vector Labs).

## Lung wet-to-dry (W/D) weight ratio

The left lung was excised, washed in PBS, gently dried on blotting paper, and weighed (wet weight). The tissue was then dried at 60°C for 72 h and weighed again (dry weight).

## Collection of bronchoalveolar lavage fluid (BALF)

BALF was collected as described (Li et al, 2013) and centrifuged at 400$g$ for 10 min to separate the supernatant from cells. The numbers of total cells, neutrophils, and macrophages from BALF were visualized using the Wright–Giemsa staining (Sigma-Aldrich) and counted in a double-blind manner. Isolation of alveolar macrophages from BALF was performed as described in detail earlier (Busch et al, 2019), and the purity of isolated macrophages was characterized by dual staining with PE-conjugated sialic acid–binding immunoglobulin-like lectin F (Siglec-F) and Alexa Fluor 488–conjugated CD11c (BioLegend) followed by flow cytometry.

## TEM

Examination of autophagosomes under TEM was performed as described earlier (Jiang et al, 2021). In short, lung tissue was cut into 1-mm$^3$ pieces and fixed in 2.5% glutaraldehyde at 4°C overnight followed by post-fixation in 1% OsO$_4$ for 3 h. After washing and dehydration in concentrated ethanol series, samples were stained with 1:1 ratio of ethanol:acetone mixture at 4°C for 20 min and then with pure acetone for a further 20 min. After embedding in resin and cutting into 70-nm sections, tissues were post-stained with uranyl acetate and lead citrate and imaged under a Hitachi H7600 TEM microscope (Hitachi).

## Cell culture and treatments

MPVECs were purchased from Creative Bioarray (Shirley) and cultured in a complete mouse endothelial cell culture medium (Creative Bioarray). To alter the expression of different targets, MPVECs were transduced with lentivirus expressing control shRNA (Lv-shNC), two distinct shRNA targeting circEXOC5 (Lv-shcir-cEXOC5#1 and #2), control overexpressing vector (OE-NC), OE-PTBP1, shPTBP1, or shRNA targeting S-phase kinase-associated protein 2

(shSkp2) (all custom produced by GenePharma). For cell treatments, at 48 h after lentiviral transduction, MPVECs were treated with LPS (100 µg/ml) without or with 3-methyladenine (3-MA; 2 mM; an autophagy inhibitor), bafilomycin A1 (2 nM; an autophagy inhibitor), or rapamycin (1 mM; an autophagy activator, all from Sigma-Aldrich). To block de novo gene transcription, MPVECs were treated with actinomycin D (10 ng/ml, Sigma-Aldrich) for indicated time periods.

## RNA FISH

Cy3-labeled circEXOC5 probe was synthesized by GenePharma. Following the instructions from a FISH assay kit (GenePharma), the probe was hybridized to cultured cells and the signal detected using a Leica TCS SP5II confocal microscope (Leica). Cell nuclei were stained with DAPI.

## Immunofluorescence

Immunofluorescence was performed on isolated macrophages from BALF cells (for TNF-α, IL-1β, and IL-6 expressions), cultured mouse cells (for detecting LC3 expression), or formalin-fixed lung tissues (for co-staining of LC3). Briefly, tissues or cells were fixed/permeabilized with cold 100% methanol for 5 min. After being washed with PBS and blocked in TBST buffer containing 2% BSA, tissues or cells were incubated with anti-TNF-α (ab183218; Abcam), anti-IL-1β (NB600-633; Novus Biologicals), anti-IL-6 (ab179570; Abcam), or anti-LC3 antibody (#83506; Cell Signaling; all diluted 1: 200) at 4°C overnight. After three washes in PBS, cells were incubated with Alexa Fluor 488– or Alexa Fluor 594–conjugated secondary antibody (Abcam) for 1 h at room temperature. For staining of LC3, the primary antibodies were mouse anti-LC3 (#83506; Cell Signaling), and the secondary antibodies were Alexa Fluor 594–conjugated secondary anti-mouse and Alexa Fluor 488–conjugated secondary anti-rabbit antibody. After another three washes, cells or tissues were with anti-fade Vectashield mounting medium with DAPI (Vector Labs) and imaged under a Leica TCS SP5II microscope.

## ELISA

The levels of TNF-α, IL-1β, and IL-6 from BALF or the cultured supernatant of target cells were measured using ELISA kits for the corresponding mouse cytokines (Abcam).

## dUTP nick-end labeling (TUNEL) assay

Cell death was examined following the protocol of a TUNEL assay kit (Solarbio), with images taken under a fluorescence microscope.

## RNA immunoprecipitation (RIP) assay

RIP assay was performed using Magna RIP RNA-Binding Protein Immunoprecipitation Kit (Millipore) according to the manufacturer's instructions. Briefly, cell lysates were prepared in RIPA buffer containing a protease inhibitor cocktail and an RNase inhibitor. Next, cell lysates were incubated with magnetic beads conjugated with anti-PTBP1 antibody (ab133734) or control normal rabbit IgG

(all from Abcam). After three washes, total RNAs were extracted from immunoprecipitates using an RNeasy mini kit (QIAGEN) and targets were examined by qRT–PCR.

### Co-immunoprecipitation (Co-IP)

Whole-cell lysates were prepared in Co-IP buffer containing 20 mM Tris–HCl, pH 8, 137 mM NaCl, 1% Nonidet P-40, 2 mM EDTA, and 1 × protease/phosphatase inhibitor cocktail (Cell Signaling). Total protein (500 µg) was incubated with 1 µg anti-Runx2 antibody (ab236639; Abcam) or rabbit IgG at 4°C for 1 h. Then, protein A-Sepharose beads (Abcam) were added to the mixture and shaken at 4°C overnight. After washing the beads and associated protein complexes three times, they were boiled in 5× sample loading buffer for 5 min and the supernatant was examined using Western blotting.

### Quantitative real-time PCR (qRT–PCR)

Total RNA from mouse tissues, whole cells, and cytoplasmic or nuclear fractions was extracted using RNA Subcellular Isolation Kit (Active Motif), and cDNA was synthesized using reverse transcriptase (Takara). RT–PCR was performed with SYBR Green Master Mix (Takara) following the manufacturer's instructions with primers. Relative gene expression was analyzed using the $2^{-\Delta\Delta Ct}$ method and normalized to that of *GAPDH* (for mRNAs) or *U6* (for circRNAs or miRNAs).

### Western blot analysis

Western blotting was performed as described previously (Zhou et al, 2018) with the following antibodies (all from Abcam): LC3I/II (ab192890; 1:1,000), Beclin1 (ab210498; 1:1,000), p62 (ab109012; 1:1,000), Runx2 (ab236639; 1:1,000), PTBP1 (ab133734; 1:1,000), Skp2 (ab183039; 1:1,000), ubiquitin (ab140601; 1:1,000), and GAPDH (ab8245; 1:5,000).

### Statistical analysis

Data were expressed as the mean ± SD from three independent in vitro experiments or from all mice of each in vivo experimental group. Statistical analyses were performed with SPSS 22.0 (IBM). Differences between experimental groups were assessed by the *t* test (for two groups) or one-way ANOVA (for more than two groups) followed by post hoc comparison. *P*< 0.05 was considered statistically significant.

## Data Availability

The datasets used or analyzed during the current study are available from the corresponding author upon reasonable request.

## Supplementary Information

## Acknowledgements

None.

### Author Contributions

P Gao: conceptualization, resources, data curation, software, formal analysis, and supervision.
B Wu: formal analysis, supervision, funding acquisition, validation, and investigation.
Y Ding: investigation, visualization, methodology, project administration, and writing—original draft.
B Yin: investigation, visualization, methodology, project administration, and writing—original draft, review, and editing.
H Gu: conceptualization, data curation, formal analysis, funding acquisition, investigation, and methodology.

### Conflict of Interest Statement

The authors declare that they have no conflict of interest.

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
