## [Reviewer comments · Life Science Alliance]

Life Science Alliance

CircEXOC5 promotes acute lung injury through the PTBP1/Skp2/Runx2 axis to activate autophagy

Pei Gao, Beirong Wu, Ding Ying, Bingru Yin, and Haoxiang Gu

DOI: <https://doi.org/10.26508/lsa.202201468>

Corresponding author(s): *Haoxiang Gu, Shanghai Children's Hospital*

Review Timeline:

Submission Date:	2022-03-31
Editorial Decision:	2022-06-17
Revision Received:	2022-09-18
Editorial Decision:	2022-10-04
Revision Received:	2022-10-09
Accepted:	2022-10-10

Scientific Editor: Novella Guidi

Transaction Report:

June 17, 2022

Re: Life Science Alliance manuscript #LSA-2022-01468

Dr. Haoxiang Gu
Shanghai Children's Hospital
No. 1400 West Beijing Road, Shanghai 200040, China.
Shanghai 200040
China

Dear Dr. Gu,

Thank you for submitting your manuscript entitled "CircEXOC5 promotes acute lung injury through the PTBP1/Skp2/Runx2 axis to activate autophagy" to Life Science Alliance. The manuscript was assessed by expert reviewers, whose comments are appended to this letter. We invite you to submit a revised manuscript addressing the Reviewer comments.

Thank you for this interesting contribution to Life Science Alliance. We are looking forward to receiving your revised manuscript.

Sincerely,

B. MANUSCRIPT ORGANIZATION AND FORMATTING:

Reviewer #1 (Comments to the Authors (Required)):

In the manuscript by Gao et al., the authors examine the role of circRNA in promoting lung injury via the autophagy pathway. The authors find that when ALI is induced in their mouse model that levels of circEXOC5 increase, which correlates with increased pathology and markers of the autophagic pathway. shRNA depletion of circEXOC5 rescues this phenotype. Mechanistically, the authors implicate the protein PTBP1 and a potential involvement of Skp2/Runx2. Overall, the data are clear and the manuscript is well written. I do have some points below that should help to make the authors conclusions more robust. I also caveat my review with the fact that I am not familiar with circRNAs or ALI models.

Main points:

- 1) The authors only use 1 shRNA sequence targeting circEXOC5. Is there potential for off target effects and should this be controlled for?
- 2) Additional higher magnification TEM images are needed in Fig. 3B and 8D. Cells look very vacuolated, which appear empty and not autophagosome-like (double/multiple limiting membranes filled with cytosolic content). Are the authors sure that they are quantifying the correct organelles here?
- 3) Lung data is consistent with increased autophagy, but it could also equally mean that autophagy is inhibited - given it is a dynamic process and markers such as LC3 and p62 can go up or down if the rate of autophagosome formation relative to the rate of degradation changes. Hence it is essential to measure autophagy flux (normally by +/- a lysosomal inhibitor). While this would be challenging to do in vivo, the authors could carry this out in MPVECs with their LPS treatments +/- Bafilomycin A1 (not chloroquine as this also activates the CASM pathway). These assays are a standard in the autophagy field and would greatly help with the robustness of the authors' conclusions.
- 4) On page 14, and in relation to Fig. 4, I think the authors are a little misleading with their modulators of autophagy. While 3-MA will inhibit autophagy, it is not a specific autophagy inhibitor. It targets PI3Kinases, which affect a multitude of cellular process. Likewise, rapamycin inhibits mTORC1, which is again involved in many autophagy-independent processes. The authors should alter their text to reflect this by saying that their results are consistent with autophagy, but other process cannot be ruled out with these compounds given the multiple roles of their targets.
- 5) In Fig.6, what is the level of overexpression of PTBP1 in these experiments? The levels relative to empty vector alone treatments would be very helpful here to determine if they are close to endogenous or vastly higher.
- 6) The data in Fig. 7 is intriguing but it is not clear what relevance Skp2/Runx2 has to ALI. Perhaps I am missing some relevant background information, as the authors do not show directly that Skp2/Runx2 levels influence the ALI phenotype, or indeed autophagy flux? Is it possible that Skp2/Runx2 could be mediating other functions of circEXOC5, which are not relevant to ALI? If so, perhaps these data could go as supplemental as potential candidates to check in future work (and change the title)?
- 7) On P17, first paragraph in relation to Fig. 7C, the authors state that simultaneous silencing of both circEXOC5 and PTBP1 fails to upregulate Skp2 - but the figure shows silencing of circEXOC5 and overexpression of PTBP1. Please clarify.

Minor points:

8) Expand abbreviations in first instance.

9) Molecular weight markers are needed on blots.

Reviewer #2 (Comments to the Authors (Required)):

The authors explore the role of a circulating RNA (CircEXOC5) in the preventing the ALI development; they describe that autophagy is a well-known mechanism involved in

ALI progress and demonstrate that circEXOC5 is a regulator of autophagy.

The authors' purpose of the study is clear. The aim is presented clearly in the manuscript and the objectives of the study are well defined. The methods used to obtain the results are right to the purpose searched. The major asset of this manuscript is that it presents new data that can be used for new therapeutic approximations. These findings may have implications for the current research.

In my point of view, the manuscript is valuable for the scientific community and the current research in ALI, but I have some comments and concerns that I feel need to be addressed to make the manuscript suitable for its publication.

Introduction:

- I miss a proper explanation of the circEXOC5 action mechanism and what is known in the literature. Please, use a paragraph to explain the circEXOC5/PTBP1/Skp2/Runx2 signaling cascade.
- As afterward, autophagy is one of the main studied pathways, authors should mention in the introduction how autophagy is affected during ALI.
- Authors mention that circEXOC5 may have a biological significance in ALI treatment, however, I disagree with this affirmation. CircEXOC5 silencing was administered long before triggering ALI, so, it will be preventing ALI development/progress, but the authors cannot confirm a therapeutic effect of it, as it was not used as a treatment after the disease was triggered. This needs to be "corrected" throughout all the manuscript and discussed in the results section. Do the authors have data regarding a therapeutic (treatment after the disease/syndrome is established) effect of CircEXOC5 or downstream pathway effectors?

Methods:

- Which parental strain of mice did the authors use? C57BL/6J or C57BL/6N? I feel it is important to mention it because both strains have different inflammatory responses under the acute stimuli.
- Why did the authors decide to use male mice? Are the data reproducible in female mice? Please, include this information in your discussion as a limitation if you only used one sex, because data are not confirmed for both sexes.
- In vivo, circEXOC5 was administered before triggering
- Please, mention the abbreviations at least once before using them (OE=overexpression, no? but it is not mentioned in the methods the first time that it is used; the same applies for Lv, MA...)

Results:

- Authors used the term normal in figure 1 for animals without damage, are they the control for the sham (CLP surgery) and vehicle mice (for the Lv injection)?
- Did the authors observe any other systemic effect due to the administration of circEXOC5? CLP-sepsis/ALI model is producing a systemic inflammation and other organs get affected due to it. Did the authors measure any circulation/systemic inflammatory marker such as IL6/IL1b in plasma? If yes, are they also mitigated by the knockdown of circEXOC5? If not, can you provide these data?
- Did the authors observe any other organ affectation due to the circEXOC5 knocking down? Any effects on the liver or kidney?
- Fig 2-C-D show isolated cells from BAF; however, these cell populations probably were quite different if isolated from an injured or non-injured animal; and they will have different responses and production of pro-inflammatory cytokines. However, the images look like the seeded cells were mainly macrophages (due to the shape). Can you indicate which cell population from BAF was used? Were they characterized? If not, how can you be sure that you are not comparing the response of a 40%macrophages: 60% neutrophil with a cell population of 90% macrophages?
- Images 5F are not clear enough to confirm if the expression is intracellular/cytoplasm/membrane. Can you please provide higher magnification pictures with less brightness that allows us to confirm your statements?
- Can you please explain/discuss for which mechanism LC3 is so strongly induced in figure 6B after shcircEXOC5 and OE-PTBP1?
- In figure 7 looks like they are shcircEXOC5 and OE-PTBP1, however in the text authors mention that after silencing both failed to upregulate skp2 (which makes more sense). Please correct the disagreement between the text and the figure.

Discussion:

- In my point of view there are missing the limitations of the study (sex of animals, prevention and not therapeutic use of circEXOC5 among others).

Response: Thanks very much for your suggestion. We provide the summary blurb as follows: 'CircEXOC5 activates autophagy and promotes acute lung injury via stimulating the Skp2 decay to stabilize Runx2 by interacting with PTBP1.'

B. MANUSCRIPT ORGANIZATION AND FORMATTING:

We encourage our authors to provide original source data, particularly uncropped/-processed electrophoretic blots and spreadsheets for the main figures of

the manuscript. If you would like to add source data, we would welcome one PDF/Excel-file per figure for this information. These files will be linked online as supplementary "Source Data" files.

Response: We appreciate the Reviewer's suggestion. Accordingly, we have uploaded original data in a Source Data file.

The original source data were shown in URL:
https://figshare.com/articles/figure/Source_Data_zip/21153658

Reviewer #1 (Comments to the Authors (Required)):

In the manuscript by Gao et al., the authors examine the role of circRNA in promoting lung injury via the autophagy pathway. The authors find that when ALI is induced in their mouse model that levels of circEXOC5 increase, which correlates with increased pathology and markers of the autophagic pathway. shRNA depletion of circEXOC5 rescues this phenotype. Mechanistically, the authors implicate the protein PTBP1 and a potential involvement of Skp2/Runx2. Overall, the data are clear and the manuscript is well written. I do have some points below that should help to make the authors conclusions more robust. I also caveat my review with the fact that I am not familiar with circRNAs or ALI models.

Main points:

1) The authors only use 1 shRNA sequence targeting circEXOC5. Is there potential for off target effects and should this be controlled for?

Response: Thanks very much for your suggestion. We originally designed two distinct shcircEXOC5 sequences (shcircEXOC5#1 and shcircEXOC5#2) and performed most experiments using both sequences. The data from both shcircEXOC5 sequences were

nearly identical and thus we only presented those from shcircEXOC5#1 in the previous draft. In the revision, we added the data from shcircEXOC5#2 into Fig. 1-5, supporting the specific biological significance of circEXOC5. Please see the revised Figures and manuscript.

2) Additional higher magnification TEM images are needed in Fig. 3B and 8D. Cells look very vacuolated, which appear empty and not autophagosome-like (double/multiple limiting membranes filled with cytosolic content). Are the authors sure that they are quantifying the correct organelles here?

Response: Thanks very much for your suggestion. We have provided Fig. 3B and 8D with TEM of higher magnification and used arrows to label autophagosomes. Please see the revised Fig. 3D and 8D.

3) Lung data is consistent with increased autophagy, but it could also equally mean that autophagy is inhibited - given it is a dynamic process and markers such as LC3 and p62 can go up or down if the rate of autophagosome formation relative to the rate of degradation changes. Hence it is essential to measure autophagy flux (normally by +/- a lysosomal inhibitor). While this would be challenging to do in vivo, the authors could carry this out in MPVECs with their LPS treatments +/- Bafilomycin A1 (not chloroquine as this also activates the CASM pathway). These assays are a standard in the autophagy field and would greatly help with the robustness of the authors' conclusions.

Response: We sincerely appreciate the Reviewer's insightful suggestion. Following your suggestion, we treated MPVECs with LSP+Lv-shcircEXOC5#1+Bafilomycin A1. As shown in updated Fig. 4, LSP+Lv-shcircEXOC5#1+Bafilomycin A1 MPVECs presented similar phenotypes as LPS+Lv-shcircEXOC5#1+3-MA in significantly suppressing the productions of pro-inflammatory cytokines (TNF- α , IL-1 β , and IL-6), alleviating apoptosis (as measured by TUNEL assay), and altering expressions of LC3, Beclin1, and p62, supporting the potency of circEXOC5 in activating autophagy. Please see the revised Fig. 4 and paragraph 4 of the Results section.

4) On page 14, and in relation to Fig. 4, I think the authors are a little misleading with their modulators of autophagy. While 3-MA will inhibit autophagy, it is not a specific autophagy inhibitor. It targets PI3Kinases, which affect a multitude of cellular process. Likewise, rapamycin inhibits mTORC1, which is again involved in many autophagy-independent processes. The authors should alter their text to reflect this by saying that their results are consistent with autophagy, but other process cannot be ruled out with these compounds given the multiple roles of their targets.

Response: Thanks very much for your suggestion. To address this comment, we have rephrased our statement and added data from using Bafilomycin A1, a specific autophagy inhibitor (see response from question 4), which support the significance of circEXOC5 in regulating autophagy. Please see paragraph 4 of the Results section.

5) In Fig.6, what is the level of overexpression of PTBP1 in these experiments? The levels relative to empty vector alone treatments would be very helpful here to determine if they are close to endogenous or vastly higher.

Response: We appreciate the reviewer's suggestion and have measured PTBP1 expression on mRNA and protein levels in OE-NC vs. OE-PTBP1 cells (Fig. 6A), which showed an approximately 3.7-fold increase of PTBP1 mRNA and close to 1.8-fold increase of PTBP1 protein. Please see the revised Fig. 6A.

6) The data in Fig. 7 is intriguing but it is not clear what relevance Skp2/Runx2 has to ALI. Perhaps I am missing some relevant background information, as the authors do not show directly that Skp2/Runx2 levels influence the ALI phenotype, or indeed autophagy flux? Is it possible that Skp2/Runx2 could be mediating other functions of circEXOC5, which are not relevant to ALI? If so, perhaps these data could go as supplemental as potential candidates to check in future work (and change the title)?

Response: We thank the reviewer for the comment. In response, we added background information on the relevance of Skp2/Runx2 to ALI (Introduction, Paragraph 3) and moved Fig. 7 as Supplementary Fig. 1. Please see the revised manuscript.

7) On P17, first paragraph in relation to Fig. 7C, the authors state that simultaneous silencing of both circEXOC5 and PTBP1 fails to upregulate Skp2 - but the figure shows silencing of circEXOC5 and overexpression of PTBP1. Please clarify.

Response: Thanks very much for your suggestion. We are sorry for the misstatement and have made correction accordingly. Please see paragraph 7 of the Results section.

Minor points:

8) Expand abbreviations in first instance.

Response: Thanks very much for your suggestion. We have double checked all abbreviations and made sure the full name was spelled out on first use in the manuscript. Please see the revised manuscript.

9) Molecular weight markers are needed on blots.

Response: Thanks very much for your suggestion. We have added molecular weight markers in all Western blot images. Please see the revised figures.

Reviewer #2 (Comments to the Authors (Required)):

The authors explore the role of a circulating RNA (CircEXOC5) in the preventing the ALI development; they describe that autophagy is a well-known mechanism involved in ALI progress and demonstrate that circEXOC5 is a regulator of autophagy.

The authors' purpose of the study is clear. The aim is presented clearly in the manuscript and the objectives of the study are well defined. The methods used to obtain the results are right to the purpose searched. The major asset of this manuscript is that it presents new data that can be used for new therapeutic approximations. These findings may have implications for the current research.

In my point of view, the manuscript is valuable for the scientific community and the current research in ALI, but I have some comments and concerns that I feel need to be addressed to make the manuscript suitable for its publication.

Introduction:

•I miss a proper explanation of the circEXOC5 action mechanism and what is known in the literature. Please, use a paragraph to explain the circEXOC5/PTBP1/Skp2/Runx2 signaling cascade.

Response: Thanks very much for your suggestion. Before this study, little is known on the circEXOC5 action mechanism in any physiological or pathological contexts. The literature search only revealed one study by Wang et al. showing that circEXOC5 promotes ferroptosis in the context of ALI. Therefore, this is the first study identifying the biological significance of circEXOC5/PTBP1/Skp2/Runx2 signaling cascade in the paradigm of ALI. We have added the background information of circEXOC5, Skp2, and Runx2, and proposed the following hypothesis: 'Therefore, we hypothesized that circEXOC5 decreased Skp2 mRNA stability by targeting PTBP1, thereby inhibiting Skp2-mediated Runx2 ubiquitin degradation in ALI.' in Paragraph 2 and 3 of the Introduction section. Please see the revised Introduction section.

•As afterward, autophagy is one of the main studied pathways, authors should mention in the introduction how autophagy is affected during ALI.

Response: Thanks very much for your suggestion. We added background information on autophagy and ALI in Paragraph 1 of the Introduction section.

•Authors mention that circEXOC5 may have a biological significance in ALI treatment, however, I disagree with this affirmation. CircEXOC5 silencing was administered long before triggering ALI, so, it will be preventing ALI development/progress, but the authors cannot confirm a therapeutic effect of it, as it was not used as a treatment after the disease was triggered. This needs to be "corrected" throughout all the manuscript and discussed in the results section. Do the authors have data regarding a therapeutic (treatment after the disease/syndrome is established) effect of CircEXOC5 or downstream pathway effectors?

Response: We thank the Reviewer for scrutinizing the manuscript for scientific stringency. We have made corrections throughout the manuscript and talked about this point as a potential caveat in the Discussion section (the second to the last paragraph).

Methods:

•Which parental strain of mice did the authors use? C57BL/6J or C57BL/6N? I feel it is important to mention it because both strains have different inflammatory responses under the acute stimuli.

Response: Thanks very much for your suggestion. C56BL/6J mice were used in this study. This information was supplemented in paragraph 1 of the Materials and Methods section.

•Why did the authors decide to use male mice? Are the data reproducible in female mice? Please, include this information in your discussion as a limitation if you only used one sex, because data are not confirmed for both sexes.

Response: Thanks very much for your suggestion. We only used male mice. Following the Reviewer's comment, we have included using male mice only as a limitation for this study in the text (Discussion, the second to the last paragraph).

•In vivo, circEXOC5 was administered before triggering

Response: Thanks very much for your suggestion. We have revised the manuscript at multiple places to suggest targeting circEXOC5 presented a preventive, instead of therapeutic effect in ALI. Please see the revised manuscript.

•Please, mention the abbreviations at least once before using them (OE=overexpression, no? but it is not mentioned in the methods the first time that it is used; the same applies for Lv, MA...)

Response: Thanks very much for your suggestion. We have double checked all abbreviations and made corrections as needed. Please see the revised manuscript.

Results:

•Authors used the term normal in figure 1 for animals without damage, are they the control for the sham (CLP surgery) and vehicle mice (for the Lv injection)?

Response: Thanks very much for your suggestion. We have revised normal to sham in Fig. 1.

•Did the authors observe any other systemic effect due to the administration of circEXOC5? CLP-sepsis/ALI model is producing a systemic inflammation and other organs get affected due to it. Did the authors measure any circulation/systemic inflammatory marker such as IL6/IL1b in plasma? If yes, are they also mitigated by the knockdown of circEXOC5? If not, can you provide these data?

Response: We appreciate the Reviewer's comment. In response, we measured the serum levels of pro-inflammatory cytokines (TNF- α , IL-1 β , and IL-6) and observed the significant impact of knocking down circEXOC5 in reducing these cytokines systemically (Fig. 2E). Please see the new Fig. 2E.

•Did the authors observe any other organ affectation due to the circEXOC5 knocking down? Any effects on the liver or kidney?

Response: We thank the Reviewer for this important comment. Unfortunately, we focused on lung injury in this study and did not examine the impacts of knocking down circEXOC5 on other organs. We have discussed this point as a potential caveat for this study (Discussion, the second to the last paragraph).

•Fig 2-C-D show isolated cells from BAF; however, these cell populations probably were quite different if isolated from an injured or non-injured animal; and they will have different responses and production of pro-inflammatory cytokines. However, the images look like the seeded cells were mainly macrophages (due to the shape). Can you indicate which cell population from BAF was used? Were they characterized? If not, how can you be sure that you are not comparing the response of a 40% macrophages: 60% neutrophil with a cell population of 90% macrophages?

Response: Thanks very much for your suggestion. We did isolate macrophages from BALF, which we provided more details in the Methods and Results section. Please see the revised manuscript.

•Images 5F are not clear enough to confirm if the expression is intracellular/cytoplasm/membrane. Can you please provide higher magnification pictures with less brightness that allows us to confirm your statements?

Response: Thanks very much for your suggestion. Following the Reviewer's suggestion, we have replaced the original Fig. 5F with images of higher magnification and improved clarity.

•Can you please explain/discuss for which mechanism LC3 is so strongly induced in figure 6B after shcircEXOC5 and OE-PTBP1?

Response: Thanks very much for your suggestion. In this study, we identified a novel signaling cascade of circEXOC5/PTBP1/Skp2/Runx2, where circEXOC5, by up-regulating and interacting with PTBP1, promoted the degradation of Skp2. Skp2 is an E3 ubiquitin ligase of Runx2; while Runx2 was shown in earlier studies to down-regulate LC3 ^[1,2] and confirmed in this study. Therefore, knocking down circEXOC5 was sufficient to suppress LPS-induced up-regulation of LC3. Because PTBP1 acts downstream of circEXOC5, overexpressing PTBP1 was able to reverse the inhibitory effect of shcircEXOC5 on LC3. We have added further explanation in the Discussion section (Paragraph 5).

[1] Qin H, Cai J. 2018. Effect of runx2 silencing on autophagy and rankl expression in osteoblasts. *Arch Oral Biol.* 95:74-78.

doi:10.1016/j.archoralbio.2018.07.016

[2] Ren C, Xu Y, Liu H, Wang Z, Ma T, Li Z, Sun L, Huang Q, Zhang K, Zhang C, et al. 2022. Effects of runt-related transcription factor 2 (runx2) on the autophagy of rapamycin-treated osteoblasts. *Bioengineered.* 13(3):5262-5276.

doi:10.1080/21655979.2022.2037881

•In figure 7 looks like they are shcircEXOC5 and OE-PTBP1, however in the text authors mention that after silencing both failed to upregulate skp2 (which makes more

sense). Please correct the disagreement between the text and the figure.

Response: Thanks very much for your suggestion. We are sorry for the typo that causes misunderstanding. The data were from simultaneously knocking down circEXOC5 and overexpressing PTBP1, where shcircEXOC5 alone up-regulated Skp2, while simultaneously knocking down circEXOC5 and overexpressing PTBP1 abolished the effect of shcircEXOC5 on Skp2, supporting the importance of PTBP1 in mediating the regulation of circEXOC5 on Skp2. We have revised the description in paragraph 7 of the Results section. Please see the revised manuscript.

.

Discussion:

- In my point of view there are missing the limitations of the study (sex of animals, prevention and not therapeutic use of circEXOC5 among others).

Response: Thanks very much for your suggestion. We have added a paragraph discussing the limitation of the study, as suggested by the Reviewer (Discussion, the second to the last paragraph). Please see the revised manuscript.

October 4, 2022

RE: Life Science Alliance Manuscript #LSA-2022-01468R

Dr. Haoxiang Gu
Shanghai Children's Hospital
No. 1400 West Beijing Road, Shanghai 200040, China.
Shanghai 200040
China

Dear Dr. Gu,

Thank you for submitting your revised manuscript entitled "CircEXOC5 promotes acute lung injury through the PTBP1/Skp2/Runx2 axis to activate autophagy". We would be happy to publish your paper in Life Science Alliance pending final revisions necessary to meet our formatting guidelines.

- please add ORCID ID for corresponding author-you should have received instructions on how to do so
- please add the Twitter handle of your host institute/organization as well as your own or/and one of the authors in our system
- please consult our manuscript preparation guidelines <https://www.life-science-alliance.org/manuscript-prep> and make sure your manuscript sections are in the correct order

Figure Check:

- Figure 3B needs scale bar

A. FINAL FILES:

- An editable version of the final text (.DOC or .DOCX) is needed for copyediting (no PDFs).

- High-resolution figure, supplementary figure and video files uploaded as individual files: See our detailed guidelines for preparing your production-ready images, <https://www.life-science-alliance.org/authors>

- Summary blurb (enter in submission system): A short text summarizing in a single sentence the study (max. 200 characters including spaces). This text is used in conjunction with the titles of papers, hence should be informative and complementary to the title. It should describe the context and significance of the findings for a general readership; it should be written in the present tense and refer to the work in the third person. Author names should not be mentioned.

B. MANUSCRIPT ORGANIZATION AND FORMATTING:

Sincerely,

Reviewer #1 (Comments to the Authors (Required)):

The authors have satisfactorily addressed my comments.

Reviewer #2 (Comments to the Authors (Required)):

The authors answered all my comments and the new revised manuscript included the information I have requested.

We would be happy to publish your paper in Life Science Alliance pending final revisions necessary to meet our formatting guidelines.

-please add ORCID ID for corresponding author-you should have received instructions on how to do so

-please add the Twitter handle of your host institute/organization as well as your own or/and one of the authors in our system

-please consult our manuscript preparation guidelines <https://www.life-science-alliance.org/manuscript-prep> and make sure your manuscript sections are in the correct order

Figure Check:

-Figure 3B needs scale bar

Response: Thanks very much for your suggestions. We have adjusted the order of the manuscript sections as required and added scale bar in Fig. 3B and Fig. 7D. Please see the revised manuscript, Fig. 3 and Fig. 7.

To upload the final version of your manuscript, please log in to your account: <https://lsa.msubmit.net/cgi-bin/main.plex>

A. FINAL FILES:

Response: Thanks very much for your suggestions. We have provided a summary blurb before the Abstract in the manuscript. Please see the revised manuscript.

B. MANUSCRIPT ORGANIZATION AND FORMATTING:

Response: Thanks very much for your suggestions. We have uploaded original data in a Source Data file. The original source data were shown in URL: https://figshare.com/articles/figure/Source_Data_zip/21153658.

****The license to publish form must be signed before your manuscript can be sent to**

production. A link to the electronic license to publish form will be sent to the corresponding author only. Please take a moment to check your funder requirements.**

October 10, 2022

RE: Life Science Alliance Manuscript #LSA-2022-01468RR

Dr. Haoxiang Gu
Shanghai Children's Hospital
No. 1400 West Beijing Road, Shanghai 200040, China.
Shanghai 200040
China

Dear Dr. Gu,

Thank you for submitting your Research Article entitled "CircEXOC5 promotes acute lung injury through the PTBP1/Skp2/Runx2 axis to activate autophagy". It is a pleasure to let you know that your manuscript is now accepted for publication in Life Science Alliance. Congratulations on this interesting work.

DISTRIBUTION OF MATERIALS:

Again, congratulations on a very nice paper. I hope you found the review process to be constructive and are pleased with how the manuscript was handled editorially. We look forward to future exciting submissions from your lab.

Sincerely,
